# DuPO: Enabling Reliable Self-Verification via Dual Preference Optimization

**Shuaijie She**[♠♡]* **Yu Bao**[♡] **Yu Lu**[♡] **Lu Xu**[♡] **Tao Li**[♡] **Wenhao Zhu**[♡]
**Jianbing Zhang**[♠] **Shujian Huang**[♠†] **Shanbo Cheng**[♡†] **Lu Lu**[♡] **Yuxuan Wang**[♡]

♠ Nanjing University ♡ ByteDance Seed

## Abstract

We present DuPO, a dual learning-based preference optimization framework that generates annotation-free feedback via generalized duality. DuPO addresses two key limitations: Reinforcement Learning with Verifiable Rewards (RLVR)'s reliance on costly labels and applicability restricted to verifiable tasks, and traditional dual learning's restriction to strictly dual task pairs (e.g., translation and back-translation). Specifically, DuPO decomposes a primal task's input into known and unknown components, then constructs its dual task to reconstruct the unknown part using the primal output and known information (e.g., reversing math solutions to recover hidden variables), broadening applicability to non-invertible tasks. The quality of this reconstruction serves as a self-supervised reward to optimize the primal task, synergizing with LLMs' ability to instantiate both tasks via a single model. Empirically, DuPO achieves substantial gains across diverse tasks: it enhances the average translation quality by 2.1 COMET over 756 directions, boosts the mathematical reasoning accuracy by an average of 6.4 points on four challenge benchmarks, and enhances performance by 9.3 points as an inference-time reranker (trading computation for accuracy). These results position DuPO as a scalable, general, and annotation-free paradigm for LLM optimization.

## 1 Introduction

Large Language Models (LLMs) (Yang et al., 2025; Grattafiori et al., 2024; Liu et al., 2024; OpenAI, 2024; Mistral AI, 2024; DeepMind, 2025) have shown remarkable progress in tasks like mathematical reasoning (Yang et al., 2024b; Shao et al., 2024; Chen et al., 2025) and multilingual translation (Cheng et al., 2025; Zhu et al., 2024; Li et al., 2024b;c). To further enhance these capabilities, researchers have increasingly adopted reinforcement learning (RL) paradigms like Reinforcement Learning from Human Feedback (RLHF) (Grattafiori et al., 2024; Liu et al., 2024; Yang et al., 2025) and Reinforcement Learning with Verifiable Rewards (RLVR) (DeepSeek-AI, 2025; Kimi Team, 2025; Yu et al., 2025; He et al., 2025; Hu et al., 2025b) have gained traction. Specifically, RLHF aligns models with human preferences but relies on costly, inconsistent human annotations (Lee et al., 2023; Zhang et al., 2024). RLVR addresses this for objective tasks (e.g., math, code) via binary rewards from verifiable answers, reducing annotation burdens. However, RLVR still depends on external supervision: acquiring verifiable answers remains a bottleneck, limiting scalability. Moreover, it struggles with generative tasks (e.g., multilingual translation), where single references cannot capture diverse high-quality outputs (Jia et al., 2025; Callison-Burch et al., 2006). Recent attempts (e.g., AI-Feedback/RLAIF (Lee et al., 2023), Constitutional AI (Bai et al., 2022)) merely swap dependencies (human labels → teacher models/rules), failing to resolve the core bottleneck.

Dual learning (He et al., 2016) offers a self-supervised alternative by leveraging *task duality* to generate intrinsic feedback: through paired "primal" and "dual" tasks (e.g., translation and back-translation (Sennrich et al., 2015)), models validate outputs via cycle consistency, eliminating reliance on external labels. Given that LLMs possess diverse capabilities from extensive pretraining, they could be trained across various tasks. However, applying this framework to LLMs is non-trivial, which faces two critical challenges:

---

†Corresponding author.
*This work was done during Shuaijie's internship at ByteDance Seed.

1. **Limited Duality in Mutually Non-Invertible Tasks**: Most real-world LLM tasks (e.g., math reasoning) lack strict invertibility. LLM's output (e.g., a math solution) rarely contains enough information to reconstruct its input (e.g., the original problem), breaking the duality cycle.

2. **Bidirectional Competence Asymmetry**: LLMs often exhibit uneven performance across primal/dual tasks (e.g., strong at solving math problems but weak at generating problems from solutions). Noisy self-signals from asymmetric tasks hinder optimization, reducing the framework's utility.

These mismatches render traditional dual learning ill-suited for general LLM optimization, leaving it an open challenge.

In this paper, we propose **DuPO** (**Du**ality-based **P**reference **O**ptimization), a framework that aligns LLM generalization with a (relaxed) duality applicable to general tasks. At its core lies a *generalized duality framework* (§3.2) built on *complementary relationships*: it decomposes each input $x$ into disjoint known ($x_k$) and unknown ($x_u$) components, then designs the dual objective to reconstruct only $x_u$ from the primal output $y$ and the known input $x_k$, rather than inverting the full input. This framework resolves two asymmetries: it restores sufficient information flow between the primal and dual tasks (task asymmetry) and reduces the complexity burden on the dual task side (capability asymmetry). The formulation naturally synergizes with LLMs: their broad foundational capabilities allow a single model to instantiate both primal and dual tasks without specific architectures, while the dual task converts the model's outputs into self-supervised reward signals, enabling continual improvement without external annotations. This bidirectional benefit addresses a critical challenge in LLM development: obtaining high-quality feedback for capability enhancement.

We empirically validate DuPO on two representative tasks: mathematical reasoning and multilingual translation, demonstrating significant and consistent improvements. By applying DuPO to one of the strongest translation LLM, Seed-X-7B-Instruct (Cheng et al., 2025), we demonstrate a significant performance gain of 2.1 COMET points on the multilingual translation benchmark, bringing the 7B model to performance comparable to ultra-large SOTA systems. In mathematical reasoning, our method yields robust gains across models of varying scales, from 1.5B to 7B parameters; notably, DuPO improves the Qwen3-4B (Yang et al., 2025) model's score on four challenging mathematical benchmarks by 6.5 percentage points. Our comprehensive ablation studies confirm that our design, the generalized duality, is crucial for achieving these results. Beyond training, DuPO acts as a *reranking mechanism* at inference, boosting performance by 9.3 points without finetuning—enabling smaller models to outperform stronger ultra-large LLMs like DeepSeek-R1 (DeepSeek-AI, 2025) even without training. In summary, DuPO reimagines task duality for non-invertible LLM tasks. It eliminates external annotation reliance, scales across tasks/domains, and enhances both training and inference, offering a scalable path to align LLMs with diverse goals using self-supervised feedback, marking a promising first step toward unlocking self-verification for broader, open-ended domains.

## 2 PRELIMINARIES

We can cast various tasks as conditional generation: with input space $\mathcal{X}$ and output space $\mathcal{Y}$, the model generates $y \in \mathcal{Y}$ via the LLM $\pi_\theta(y \mid x)$ given an input $x \in \mathcal{X}$.

### 2.1 PREFERENCE OPTIMIZATION

Preference optimization steers LLMs' behavior by assigning scalar rewards to responses based on their quality, higher-rewarded outputs are preferred and reinforced by the optimization. Formally, given a reward function $r : \mathcal{X} \times \mathcal{Y} \to \mathbb{R}$ that quantifies output quality, the objective is:

$$\max_\theta \mathbb{E}_{x \sim \mathcal{D}, y \sim \pi_\theta(\cdot \mid x)}[r(x, y)] \tag{1}$$

In practice, $r(x, y)$ is commonly derived from human preferences (RLHF), LLM judgments (RLAIF), or verifiable correctness (RLVR). While numerous effective algorithms (e.g., PPO (Schulman et al., 2017), REINFORCE++ (Hu et al., 2025a), GRPO (Shao et al., 2024)) have been developed to optimize this objective, their performance ultimately hinges on the quality of the reward signal. Consequently, the key challenge lies in obtaining accurate and scalable rewards, given that existing sources are plagued by high costs, inherent biases, and limited coverage.

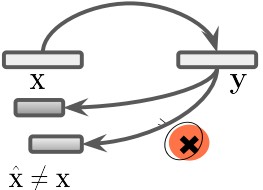

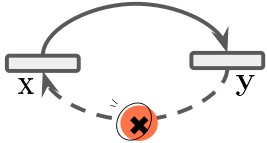

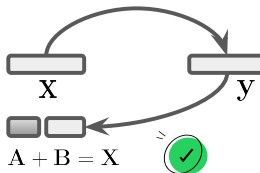

(a) Non-unique reconstruction of $\mathbf{x}$ from $\mathbf{y}$

(b) Failure to reconstruct $\mathbf{x}$ given $\mathbf{y}$

(c) Predicting a subcomponent $\mathbf{B}$ of the input to preserve uniqueness and adapt dual task difficulty

Figure 1: **Challenges in Dual Learning and Solutions via Relaxed Duality Constraints.** Dilemmas in applying dual learning — (a) Non-unique reconstruction of $\mathbf{x}$ from $\mathbf{y}$ breaks the closed-loop; (b) Failure to reconstruct $\mathbf{x}$ from $\mathbf{y}$ due to competence asymmetry. Resolutions by relaxing duality restrictions — (c) Predicting unknown component $\mathbf{B}$ to preserve uniqueness and adapt dual task difficulty

## 2.2 DUAL LEARNING

Dual learning offers a self-supervision signal by utilizing the task duality. We begin by formalizing the task duality between a *primal* task and its *dual* counterpart.

**Definition 1.** *A primal task is a mapping $\mathcal{T}_p : \mathcal{X} \to \mathcal{Y}$, and a dual task is a mapping $\mathcal{T}_d : \mathcal{Y} \to \mathcal{X}$. The pair $(\mathcal{T}_p, \mathcal{T}_d)$ is said to form a dual pair if they satisfy the consistency principle:*

$$\forall \mathbf{x} \in \mathcal{X}, \quad d\big(\mathbf{x}, \mathcal{T}_d(\mathcal{T}_p(\mathbf{x}))\big) \leq \epsilon_{\mathcal{X}},$$

*where $d(\cdot) : \mathcal{X} \times \mathcal{X} \to \mathbb{R}^+$ is a domain-specific distance metric, and $\epsilon_{\mathcal{X}} \geq 0$ is a tolerance threshold that quantifies acceptable reconstruction errors in each space.*

Leveraging this duality, we can construct a self-supervised reward to quantify the quality of a primal-task output. Given an input $\mathbf{x}$ and its corresponding output $\mathbf{y} = \mathcal{T}_p(\mathbf{x})$, we could define reward as

$$r(\mathbf{x}, \mathbf{y}) \propto \exp\big(-\lambda \cdot d\big(\mathbf{x}, \mathcal{T}_d(\mathbf{y})\big)\big), \tag{2}$$

where $\lambda > 0$ controls the sensitivity of the reward to reconstruction error. High-quality outputs maximize the expected reward $\mathbb{E}[r(\mathbf{y})]$ by preserving information that is recoverable through the duality cycle. This principle has been successfully applied in various domains, including machine translation (He et al., 2016; Zou et al., 2025).

## 3 DUAL LEARNING-BASED PREFERENCE OPTIMIZATION FRAMEWORK

In this section, we present **DuPO** (**Du**ality-based **P**reference **O**ptimization). Its core objective is to leverage the intrinsic relationships between tasks and their dual counterparts to generate self-supervised rewards, enabling LLMs to improve performance without relying on expensive human annotations or complex handcrafted rules.

### 3.1 CHALLENGES IN DUAL LEARNING-BASED OPTIMIZATION

While task duality offers a promising self-supervised paradigm, its application to LLM optimization is non-trivial, as it confronts two critical challenges that disrupt the closed-loop information flow.

**Challenge I: Limited Duality in Non-Mutually Implicative Tasks.** The utility of task duality hinges on $\mathcal{T}_p$ and $\mathcal{T}_d$ being mutually implicative — specifically, the output $\mathbf{y}$ of $\mathcal{T}_p$ contains sufficient information to reconstruct $\mathbf{x}$ via $\mathcal{T}_d$, and vice versa. This property holds for canonical tasks like machine translation, where $\mathbf{y}$ (a translation) preserves the semantic content of $\mathbf{x}$ (the source sentence), enabling $\mathcal{T}_d$ (back-translation) to recover $\hat{\mathbf{x}} \approx \mathbf{x}$.

However, most real-world tasks lack this mutual implicativity (Fig. 1a). Consider mathematical reasoning, where $\mathcal{T}_p$ solves a problem $\mathbf{x}$ (e.g., "A box contains 3 red and 5 blue balls; what is the total?") to produce $\mathbf{y} = 8$. Here, $\mathbf{y}$ (the total count) is insufficient to uniquely reconstruct $\mathbf{x}$ via $\mathcal{T}_d$, as 8 could answer infinitely many disparate questions, such as "What is $10 - 2$?" or "What is the

atomic number of Oxygen?". This underdetermined relationship breaks the duality loop: $\mathcal{T}_d$ cannot reliably recover $\mathbf{x}$ from $\mathbf{y}$, making the self-supervised reward (based on $\hat{\mathbf{x}} \approx \mathbf{x}$) untrustworthy.

**Challenge II: Bidirectional Competence Asymmetry.** Even for mutual invertibility tasks, duality optimization is sensitive to the bidirectional competence of the LLM — the performance gap between $\mathcal{T}_p$ and $\mathcal{T}_d$. If $\mathcal{T}_p$ is strong but $\mathcal{T}_d$ is weak, $\mathcal{T}_d$ may produce noisy $\hat{\mathbf{x}}$ that distorts the supervision signal (Fig. 1b). This asymmetry is particularly pronounced in LLMs, where extensive pretraining creates diverse but uneven capabilities across tasks, even within the same domain.

For instance, in machine translation optimization, let $\mathbf{x}$ = "The quick brown fox jumps over the lazy dog" (English) and $\mathbf{y}$ = "Der schnelle braune Fuchs springt über den faulen Hund" (correct German translation). A high-quality $\mathbf{y}$ should enable $\mathcal{T}_d$ to back-translate to $\hat{\mathbf{x}} \approx \mathbf{x}$. However, if $\mathcal{T}_d$ struggles with nuanced vocabulary (e.g., "schnelle" → "fast" instead of "quick"), $\hat{\mathbf{x}}$ might be "The fast brown fox jumps over the lazy dog" — a divergence from $\mathbf{x}$ that erroneously penalizes $\mathbf{y}$ despite its correctness.

Using separate models for $\mathcal{T}_p$ and $\mathcal{T}_d$, as was common in traditional dual learning, merely sidesteps the challenge of intrinsic competence asymmetry (Wang et al., 2018). This imbalance still arises from the distinct natures and complexities of the primal-dual tasks, destabilizing the self-supervised feedback loop.

## 3.2 GENERALIZED DUALITY REWARD VIA COMPLEMENTARY TASK

To address the two-fold challenges, we propose *generalized duality* that redefines task duality through complementary dependencies. It transcends traditional duality's strict input-output reversal requirement by leveraging *partial and stable dependencies* between task components, enabling robust self-supervised rewarding even for tasks lacking inherent mutual implicativity.

**Definition 2.** *Let the input space $\mathcal{X}$ of a primal task $\mathcal{T}_p$ be decomposed into two disjoint subspaces: $\mathcal{X}_k$ (known components) and $\mathcal{X}_u$ (unknown components), such that $\mathcal{X} = \mathcal{X}_k \times \mathcal{X}_u$. The primal task $\mathcal{T}_p$ is a mapping $\mathcal{T}_p : \mathcal{X} \to \mathcal{Y}$ that maps $\mathbf{x} \in \mathcal{X}$ to an output space $\mathbf{y} \in \mathcal{Y}$. Its **complementary dual task** $\mathcal{T}_{cd}$ is a mapping that leverages $\mathbf{y}$ and the known component $\mathbf{x}_k$ to reconstruct the unknown component $\hat{\mathbf{x}}_u \in \mathcal{X}_u$:*

$$\mathcal{T}_{cd} : (\mathbf{y}, \mathbf{x}_k) \mapsto \hat{\mathbf{x}}_u.$$

*The pair $(\mathcal{T}_p, \mathcal{T}_{cd})$ is said to form a generalized dual pair if they satisfy the complementary consistency principle:*

$$\forall \mathbf{x} \in \mathcal{X}, \ \mathbf{y} = \mathcal{T}_p(\mathbf{x}) : \ d\big(\mathbf{x}_u, \mathcal{T}_{cd}(\mathbf{y}, \mathbf{x}_k)\big) \leq \epsilon,$$

*where $d(\cdot) : \mathcal{X}_u \times \mathcal{X}_u \to \mathbb{R}_+$ is a domain-specific distance metric, and $\epsilon \geq 0$ is a tolerance threshold.*

Leveraging this generalized duality, we can construct a self-supervised reward to quantify the preference of a primal-task output analogously to Def. 1. Given an input $\mathbf{x} \in \mathcal{X}$ with decomposition $x = (x_k, x_u)$ and its corresponding output $y = \mathcal{T}_p(x)$, the reward is defined as

$$r(\mathbf{x}, \mathbf{y}) \propto \exp\left(-\lambda \cdot d\left(\mathbf{x}_u, \mathcal{T}_{cd}(\mathbf{y}, \mathbf{x}_k)\right)\right), \tag{3}$$

where $\lambda > 0$ controls reward sensitivity.

A simple two-sum task is illustrated in Example 1. When generalizing to practical applications, DuPO also splits the input and selects a component of the input (e.g., a numerical variable in math problem) to serve as the unknown part $x_u$ and construct the corresponding dual task. To further improve the task duality, the selection could follow principles like *Answerability of the Dual Question* and *Uniqueness of the Correct Completeness* (see Appendix A for details). Moreover, DuPO's flexibility also allows for task-specific distance metrics $d(\cdot)$. We could employ BLEU scores for multilingual translation, while for mathematical reasoning, we evaluate variable equality, yielding binary rewards, with case studies provided in Appendix D.

---

**Example 1: Generalized Duality Reward for a Two-Sum Task:** $A + B$

The primal task $\mathcal{T}_p : \mathbf{y} \leftarrowtail \mathbf{x}_u + \mathbf{x}_k$ is to compute the sum of two numbers, with its input and output as:

– The input $\mathbf{x}$ is decomposed as $\mathbf{x} \leftarrowtail (A, B)$, where $\mathbf{x}_k = A$ (a known number) and $\mathbf{x}_u = B$ (an unknown number, without loss of generality).

– The output $\mathbf{y}$ is the result of sum: $C = A + B$.

The complementary dual task $\mathcal{T}_{cd} : \mathbf{x}_u \leftarrowtail \mathbf{y} - \mathbf{x}_k$ is designed to reconstruct the unknown component $\mathbf{x}_u$, using the primal output $\mathbf{y}$ (i.e. $C$) and the known $\mathbf{x}_k$ (i.e. $A$):

$$\hat{\mathbf{x}}_u \leftarrowtail B' = C - A$$

Then, we can directly quantify whether $B$ (original unknown) and $B'$ (reconstructed unknown) are consistent as reward signal:

$$r(\mathbf{x}, \mathbf{y}) \propto \exp\left(-\lambda \cdot \mathbb{I}(B \neq B')\right).$$

Here, $\mathbb{I}(\cdot)$ is an indicator function: it equals 0 if $B = B'$ (consistent) and 1 otherwise (inconsistent). This ensures the reward is maximized when $B$ and $B'$ match, and reduced otherwise.

---

### 3.3 Policy Optimization with Generalized Duality Reward

The core of our Dual Learning-based Policy Optimization (DuPO) framework is to optimize LLMs using duality-derived self-supervised complementary rewards $r(\mathbf{x}, \mathbf{y})$, without external annotations. The objective is to maximize the expected reward based on the (complementary) dual task:

$$\mathcal{J}(\theta) = \mathbb{E}_{\mathbf{y} \sim \pi_\theta(\mathbf{y}|\mathbf{x})}\left[r(\mathbf{x}, \mathbf{y})\right], \tag{4}$$

where $\pi_\theta(\mathbf{y}|\mathbf{x})$ denotes the LLM (parameterized by $\theta$) for generating output $\mathbf{y}$ given input $\mathbf{x} = (\mathbf{x}_u, \mathbf{x}_k)$. Notably, DuPO is compatible with various RL algorithms (e.g., PPO, REINFORCE++), we adopt GRPO (Shao et al., 2024) in our experiments for its simplicity and efficiency.

**Remark 1.** *Compared to traditional dual learning, which suffers from strict mutual implicativity (i.e., $\mathbf{y}$ must fully encode $\mathbf{x}$) and bidirectional competence asymmetry, our generalized duality framework offers three fundamental advantages:*

1. ***Overcomes the Invertibility Constraint.*** *By redesigning the dual objective from reconstructing the full input $\mathbf{x}$ to only a selected unknown component $\mathbf{x}_u$, our framework fundamentally bypasses the stringent requirement of task symmetry. This relaxation is the key to unlocking dual learning for tasks that are inherently non-invertible, where the primal output does not contain sufficient information to recover the entire input.*

2. ***Mitigates the Competence Asymmetry.*** *The difficulty of the dual task is significantly reduced in two ways. First, the known component $\mathbf{x}_k$ acts as a strong contextual anchor, constraining the solution space for reconstruction. Second, we can simply yet effectively select an $\mathbf{x}_u$ that is not only feasibly reconstructible but also acts as a faithful reward signal for the primal task's solution quality (Appendix A). This directly addresses the "weak dual" pitfall and ensures the self-supervised reward is reliable and informative.*

3. ***Enables Broad Applicability.*** *It unlocks dual learning for a broad class of tasks previously considered unsuitable, including complex domains such as mathematical reasoning, code generation, and dialogue systems where input-output relationships are partial.*

## 4 Experiment

We validate the efficacy of DuPO on two representative tasks: multilingual translation and mathematical reasoning. Below, we detail the experimental setup, datasets, and evaluation metrics for each task, followed by key results.

### 4.1 Experiment Setup

**Base Model.** We evaluate DuPO on a diverse set of strong and popular base models to demonstrate its effectiveness and robustness. For translation tasks, we employ Seed-X-7B-Instruct (Cheng et al., 2025), one of the strongest open-source translation models. For mathematical reasoning, we select models of varying scales and capabilities, including small-scale yet powerful DeepSeek-R1-Distill-Qwen-1.5B (DeepSeek-AI, 2025) and its larger counterpart DeepSeek-R1-Distill-Qwen-7B, both

| Model | BLEU | COMET | BLEURT | Avg. |
|-------|------|-------|--------|------|
| Qwen3-8B | 21.7 | 84.8 | 65.8 | 57.4 |
| Doubao-1.5-Thinking | 26.2 | 87.9 | 71.7 | 61.9 |
| Qwen3-235B-22B | 28.4 | 88.8 | 73.9 | 63.7 |
| DeepSeek-R1-0528 | 30.2 | 89.2 | 75.0 | 64.8 |
| Seed-X-7B-Instruct | 28.8 | 87.0 | 72.6 | 62.8 |
| w/ DuPO (ours) | 30.3 | 89.1 | 74.6 | 64.7 |

Table 1: **Multilingual Translation Performance Across** 756 **Translation Directions in** 28 **Languages.** DuPO significantly improves all metrics and performs comparably to its strong counterparts (DeepSeek models).

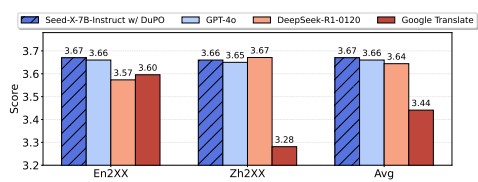

Figure 2: **Human Evaluation Scores (0-4) on the Seed-X-Challenge for 14 Language Directions.** DuPO achieves performance comparable to or even surpassing ultra-large models such as GPT-4o and DeepSeek-R1-0120, while significantly outperforming Google Translate.

distilled from the SOTA DeepSeek-R1. We also include Qwen3-4B (Yang et al., 2025), the latest strong small LLM, and the most capable open-source reasoning model, OpenReasoning-Nemotron-7B (Moshkov et al., 2025). These models represent strong and representative baselines within their respective model scales, ensuring comprehensive evaluation. Additionally, we also include some SOTA and impressive ultra-large models like Doubao-1.5/1.6-Thinking (ByteDance Seed Team, 2025), Claude-Sonnet4-Thinking, and DeepSeek-R1 (DeepSeek-AI, 2025) for comparison.

**Dataset.** For translation tasks, we focus on 28 languages that are aligned with the language coverage of Seed-X, selecting 1,000 prompts for each language from a multilingual pre-training dataset to create our training prompt set. Additionally, we collect 7,000 parallel data entries across these specified languages to support our experiments from the dev set of Flores-200 (NLLB Team, 2024). For mathematical reasoning tasks, we utilize questions from a mixture of publicly available mathematics datasets[*]. These diverse datasets are widely used to synthesize SFT data or provide oracle labels for RL, training LLMs on complex reasoning tasks.

**Benchmarks.** For comprehensive evaluation, we include various tasks and benchmarks:

- **Multilingual Translation:** We construct an automatic evaluation set by randomly sampling 50 instances for each of the 756 translation directions (across 28 languages) from the Flores test set[†]. This dataset, comprising 37,800 samples, will be released to facilitate reproducibility. We employ BLEU (Papineni et al., 2002), COMET (Rei et al., 2020), and BLEURT (Sellam et al., 2020) as evaluation metrics. For human evaluation, we use the Seed-X-Challenge (Cheng et al., 2025)[‡], a challenging benchmark to test the boundaries of LLMs' translation capabilities with diverse linguistic elements across multiple domains. Human experts assess accuracy, fluency, and idiomaticity, scoring translations from Chinese or English to seven languages on a 0-4 scale (higher score denotes better translation).

- **Mathematical Reasoning:** We evaluate our approach on multiple benchmarks, including AMC23 (MAA, 2023), AIME24, AIME25 and HMMT (Balunović et al., 2025), to assess performance in standardized contest environments. For each problem, we sample 32 responses using a temperature of 0.8 and a maximum reasoning budget of 32,000 tokens, then report the average accuracy (Avg@32).

Ultra-large models like DeepSeek-R1 and Doubao-1.6-Thinking are accessed via their official APIs. More details about training are provided in Appendix C.

## 4.2 MAIN RESULTS

### 4.2.1 DuPO BOOSTS LLM'S PERFORMANCE ON VARIOUS TASKS

DuPO achieves strong performance on diverse tasks, including multilingual translation and mathematical reasoning. On multilingual translation, DuPO elevates the backbone to a SOTA performance level, rivaling and even surpassing significantly ultra-large LLMs. As detailed in Tab. 1, applying DuPO to the Seed-X-7B-Instruct model boosts its performance by 1.5, 2.1, and 2.0 across three automatic evaluation metrics. This performance even surpasses that of current SOTA closed-source

---

[*]More details on math data preparation can be found in Appendix B.

[†]https://huggingface.co/datasets/openlanguagedata/flores_plus

[‡]https://github.com/ByteDance-Seed/Seed-X-7B/tree/main/challenge_set

| Model | AMC23 | AIME24 | AIME25 | HMMT | Avg. |
|---|---|---|---|---|---|
| DeepSeek-R1-0120 | 97.7 | 79.8 | 70.0 | 44.2 | 72.9 |
| Claude-Sonnet4-Thinking | 97.5 | 82.5 | 70.0 | 48.3 | 74.6 |
| Doubao-1.5-Thinking | 99.4 | 86.3 | 73.3 | 57.7 | 79.2 |
| Doubao-1.6-Thinking | 98.8 | 88.4 | 83.4 | 60.1 | 82.7 |
| DeepSeek-R1-0528 | 99.4 | 91.4 | 87.5 | 71.4 | 87.4 |
| DeepSeek-R1-Distill-Qwen-1.5B | 67.5 | 20.0 | 20.0 | 13.3 | 30.2 |
| **w/ DuPO (ours)** | 72.5 | 30.0 | 26.7 | 16.7 | 36.5 (+6.3) |
| DeepSeek-R1-Distill-Qwen-7B | 85.0 | 56.7 | 36.7 | 20.0 | 49.6 |
| **w/ DuPO (ours)** | 90.0 | 63.3 | 40.0 | 26.7 | 55.0 (+5.4) |
| Qwen3-4B | 95.0 | 70.0 | 66.7 | 40.0 | 67.9 |
| **w/ DuPO (ours)** | 97.5 | 83.3 | 70.0 | 46.7 | 74.4 (+6.5) |
| OpenReasoning-Nemotron-7B | 95.0 | 83.3 | 73.3 | 56.7 | 77.1 |
| **w/ DuPO (ours)** | 97.5 | 83.3 | 90.0 | 66.7 | 84.4 (+7.3) |

Table 2: **Mathematical Reasoning Performances (%) on Representative Benchmarks.** DuPO significantly improves the performances across models with varying base capabilities, enabling Qwen3-4B to outperform DeepSeek-R1-0120 and OpenReasoning-Nemotron-7B to achieve impressive performance (+7.3).

ultra-large language models, such as Doubao-1.5-Thinking (+2.8) and Qwen3-235B-22B (+1.0), and is on par with the performance of the latest DeepSeek-R1. As shown in Fig. 2, DuPO demonstrates remarkable performance, achieving results comparable to state-of-the-art ultra-large models such as GPT-4o and DeepSeek-R1. Moreover, DuPO substantially outperforms widely-used commercial closed-source systems like Google Translate, showcasing a clear advantage in translation quality as perceived by human evaluators.

On mathematical reasoning, the results in Tab. 2 clearly demonstrate that DuPO yields consistent and significant performance improvements across all models at different scales and baseline reasoning ability. On the most powerful OpenReasoning-Nemotron-7B model, applying DuPO increased the average score from 77.1% to 84.4%, achieving impressive performance. This trend of significant gains continues on the mid-sized Qwen3-4B model, which saw its average score boosted by 6.5 points from 67.9% to 74.4%, even surpassing the ultra-large model DeepSeek-R1-0120. The approach remains remarkably effective on DeepSeek's distilled models as well. Even on DeepSeek-R1-Distill-Qwen-1.5B, the least reasoning capability among the strong baselines, we still achieved a 6.3-point increase in average accuracy. Our framework's performance is further validated by concrete examples in multilingual translation and mathematical reasoning (case studies in Appendix D).

### 4.2.2 DuPO Scales to Various Backbones Effectively

To validate the robustness and generalization of our proposed DuPO framework, we extend our evaluation to the LlaMA architectural family. Our experiments are conducted on two LlaMA architectural models: LlaMA-3.1-8B (Grattafiori et al., 2024) and OctoThinker-8B-Hybrid-Base (Wang et al., 2025), the latter of which has undergone middle training on mathematical reasoning knowledge. Considering the significant difference in model ability, we select two benchmarks of moderate difficulty, AMC23 (MAA, 2023) and MATH500 (Hendrycks et al., 2021). For a fair comparison, all models are finetuned using identical training data and settings. Results are listed in Tab. 3.

As seen, DuPO's effectiveness is not tied to a specific model architecture; it serves as a robust and generalizable enhancement, delivering significant improvements to diverse backbones regardless of their initial reasoning proficiency. DuPO lifts the average score of LlaMA-3.1-8B to 32.1%, a +24.0 percentage-point gain over the vanilla model, and surpasses SimpleRL-Zoo (Zeng et al., 2025) (which relies on oracle-labeled answers during training) by 13.1%. When applied to the OctoThinker-8B-Hybrid-Base (Wang et al., 2025), our DuPO approach yields even more impressive performance improvements of +50.0 on AMC23 and +27.4 on MATH500, achieving an average performance of 62.5.

| Model | AMC23 | MATH500 | Avg. |
|---|---|---|---|
| LlaMA-3.1-8B | 2.5 | 13.6 | 8.1 |
| **w/ SimpleRL-Zoo** | 15.0 | 23.0 | 19.0 |
| **w/ DuPO (ours)** | 20.0 | 44.2 | 32.1 |
| OctoThinker-8B-Hybrid-Base | 5.0 | 42.6 | 23.8 |
| **w/ DuPO (ours)** | 55.0 | 70.0 | 62.5 |

Table 3: **Performances (%) of DuPO on Different Backbone Models.** DuPO even surpasses SimpleRL-Zoo, which utilizes labeled answers as reward. DuPO's potential is further exemplified by OctoThinker, which underwent additional mid-training.

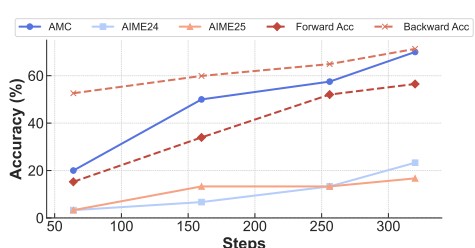

Figure 3: **Training Progress of DuPO on Qwen3-4B-Base.** The performance consistently improves on the primal/dual task (Forward/Backward Acc) and the benchmarks.

| Model | AIME24 | AIME25 | Avg. |
|---|---|---|---|
| DeepSeek-R1-0120 | 79.8 | 70.0 | 74.9 |
| Claude-Sonnet4-Thinking | 82.5 | 70.0 | 76.3 |
| DeepSeek-R1-Distill-1.5B | 20.0 | 20.0 | 20.0 |
| **w/ DuPO rewarding** | 53.3 | 24.1 | 38.7 (+18.7) |
| Qwen3-4B | 70.0 | 66.7 | 68.4 |
| **w/ DuPO rewarding** | 86.6 | 68.9 | 77.7 (+9.3) |

Table 4: **Inference-Time Scaling on Mathematical Reasoning Using DuPO Rewarding (Backward Acc) for Reranking**. Our method improves the performance of policy models with varying base ability, without requiring additional training.

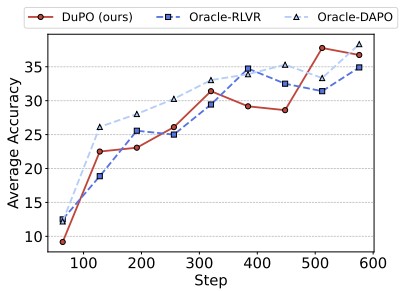

Figure 4: **Training Process of DuPO vs. Oracle Baselines.** DuPO tracks the oracle baselines that uses ground-truth for verifiable reward.

### 4.2.3 DuPO INCENTIVIZES REASONING CAPABILITY ON BASE MODEL

Further validating DuPO's versatility, we demonstrate its effectiveness even when applied directly to a base model without a preliminary supervised fine-tuning (SFT) phase for activation of reasoning behavior. We track the learning dynamics by simultaneously collecting the primal task accuracy ("Forward Acc") and complementary dual task accuracy ("Backward Acc") on the training set and its generalization performance on three test set.

As shown in Fig. 3, DuPO provides a stable and effective pathway to awaken and generalize the latent reasoning abilities of a base model, validating its utility as a powerful training methodology. Specifically, the training dynamics reveal a clear and substantial improvement on the primal task, with the "Forward Acc" soaring from a nascent 15.2% to 56.5%. Our automated unknown-component selection strategy (based on Qwen3-4B-Instruct) adjusts dual-question difficulty, yielding relatively high initial accuracy (52.6%) and effective rewards. As the model improves during training, it naturally solves more dual questions, thereby unlocking richer reward signals. More importantly, this acquired skill demonstrates robust generalization. Performance on the test set AMC23 leaped from 20% to 70%, with similarly significant gains observed on the AIME24 and AIME25 datasets.

### 4.2.4 DuPO ACHIEVES SIMILAR PERFORMANCE WITH ORACLE

Further analysis reveals that DuPO achieves near-optimal performance, effectively approaching the results obtainable with ground-truth supervision. To quantify this gap, we construct two oracle baselines, Oracle-RLVR and Oracle-DAPO, where both utilize ground-truth answers to verify rollouts as reward signals. Specifically, Oracle-RLVR utilizes the exact same dataset as DuPO. In contrast, Oracle-DAPO employs the DAPO Yu et al. (2025) dataset, which is a representative high-quality annotated dataset from the open-source community. These oracles indicate the performance upper bound of the RLVR paradigm given accurate supervision. For efficiency, we conducted experiments using Qwen3-4B-Base with 8192 maximum output length.

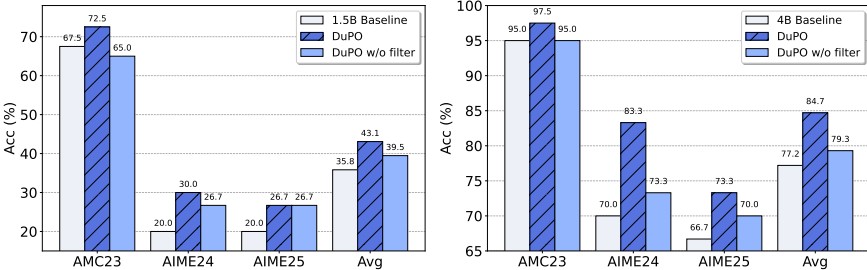

Figure 5: **Performance Ablation of DeepSeek-R1-Distill-Qwen-1.5B/Qwen3-4B on Mathematical Reasoning**. Our unknown component selection strategy reduces training noise and improves these models' performance across three benchmarks.

As shown in Figure 4, DuPO closely tracks Oracle-RLVR throughout the training process. By step 600, both methods reach nearly identical performance levels at approximately 35% accuracy. This close alignment indicates that our self-verification reward signals are accurate and effective. Furthermore, we observe that the performance of Oracle-DAPO is comparable to that of both DuPO and Oracle-RLVR. This similarity suggests that our $x_u$ selection strategy has a minimal impact on the data distribution, thereby demonstrating the robustness of DuPO.

### 4.2.5 DuPO Scales Reasoning during Inference without Training

Beyond serving as a reward signal for RL training, the DuPO mechanism can be naturally applied as a training-free, inference-time reranking strategy to improve the reasoning capabilities of any LLM. The process unfolds in three stages: 1) Similar to the rollout stage during RL process, we prompt any given policy model to generate diverse reasoning trajectories. 2) For each candidate trajectory, we use its final answer to ask the policy model to solve the corresponding dual question automatically constructed without accessing labeled answer. We could apply more computation by performing $K$ ($K = 8$ in our experiments) sampling runs on each dual question for a more reliable reward estimate, a practice distinct from RL training. 3) Finally, for each test set question, we select the trajectory with the highest backward accuracy on its dual questions as the final output.

As presented in Tab. 4, the experimental results demonstrate that DuPO provides accurate reward signals, effectively guiding models towards correct reasoning, serving as an efficient approach for scaling reasoning capabilities even without training. On the two challenging AIME benchmarks, applying DuPO as a reranking method improves the average performance of Qwen3-4B by 9.3 points, elevating its accuracy from 68.4% to 77.7% without any additional training. Notably, the DuPO-enhanced Qwen3-4B surpasses DeepSeek-R1 and Claude-Sonnet4-Thinking (77.7% vs. 74.9%/76.3% on average). The impact on DeepSeek-R1-Distill-Qwen-1.5B is even more pronounced, with an 18.7 point increase (20.0% to 38.7%).

### 4.2.6 Effects of Task Duality

To verify the efficacy of our component selection strategy, we conduct an ablation study by removing it and training on the entire unfiltered dataset. The results, illustrated in Figure 5, reveal a clear trend: removing the selection strategy results in a significant performance degradation across most benchmarks. This is further evidenced by an average accuracy drop of 3.6 and 5.4 percentage points for the 1.5B and 4B models, respectively. The result validates our strategy's effectiveness, confirming that by improving task duality, it provides a cleaner reward signal that is crucial for achieving superior performance.

## 5 Related Work

### 5.1 Preference Optimization for LLMs

Preference optimization is pivotal for aligning LLMs with desired behaviors, with current research dominated by three paradigms reliant on external supervision. RLHF (Ouyang et al., 2022) aligns models with human preferences, but is fundamentally hampered by the cost and inconsistency of human annotation (Lee et al., 2023; Zhang et al., 2024). As a cost-effective alternative, LLM-as-a-Judge (Zheng et al., 2023; Lee et al., 2023) utilizes a powerful LLM for evaluation. The reliability

is perpetually undermined by the judge model's own performance limitations and intrinsic biases, such as sensitivity to response ordering or stylistic artifacts (Wang et al., 2024; Gudibande et al., 2023; Li et al., 2024a). In parallel, RLVR has shown success in domains like mathematics by using ground-truth outcomes as reward signals (DeepSeek-AI, 2025; Kimi Team, 2025; Yang et al., 2025). However, this paradigm's continued reliance on labeled answers as external supervision makes it ill-suited for free-form tasks, such as multilingual translation, that inherently lack a single, definitive ground truth. Recent self-supervised paradigms explore alternative reward sources. Some leverage consistency checks between problem paraphrases (Zhang et al., 2025), while others utilize self-play to generate a curriculum (Huang et al., 2025), which requires managing a multi-agent adversarial process. RLT (Cetin et al., 2025) focuses on training a teacher model to generate more effective distillation data, by rewarding the teacher's explanations based on the student's performance. DuPO differs by sourcing its reward from the intrinsic, dual structure of a task itself and provides a reliable and self-contained verification signal, sidestepping dependencies on auxiliary data generation or the complexities of adversarial dynamics.

## 5.2 DUAL LEARNING

Dual learning enhances model performance by leveraging intrinsic task symmetry, where a primal task and its complementary dual task mutually provide supervision. He et al. (2016) first introduced dual learning for machine translation, which uses bidirectional tasks (e.g., En→Zh and Zh→En) to generate pseudo-labels via back-translation (Sennrich et al., 2015), reducing reliance on parallel corpora—a breakthrough for low-resource language pairs. Building on this foundation, the paradigm has proven highly versatile—spanning multi-modal (Yi et al., 2017; Zhu et al., 2017; Ren et al., 2020) and knowledge reasoning (Dognin et al., 2020), and extending to reinforcement learning (Luo et al., 2019; Bahng et al., 2025). In modern LLMs, it further refines output quality and enforces semantic consistency (Zou et al., 2025; Chen et al., 2024). However, the reliance on strict task duality, requiring mutually invertible tasks—precludes its application to open-ended or creative domains. In this work, we generalize the dual learning paradigm, moving beyond this rigid invertibility constraint.

## 6 CONCLUSION

We introduce DuPO, a dual learning-based preference optimization framework that eliminates the need for costly human annotations and handcrafted rewards in LLM training. Its core innovation, *generalized duality*, generates self-supervised feedback by decomposing and reconstructing input spaces, addressing critical limitations of traditional dual learning and preference optimization paradigms. DuPO's effectiveness is validated across diverse tasks: in mathematical reasoning, it boosts average accuracy by 6.4 percent points across models from 1.5B to 7B, while in multilingual translation, it elevates a 7B model to rival larger SOTA LLMs with COMET score gains of up to 2.1 across 756 translation directions. Furthermore, as a training-free reranker, DuPO enables smaller models to outperform significantly larger LLMs by up to 9.3 points. This model-agnostic and task-versatile design positions DuPO as a scalable, annotation-efficient solution for more autonomous, adaptable, and cost-effective LLM optimization.

## REPRODUCIBILITY STATEMENT

To ensure the full reproducibility of our work, we detail the step-by-step application of our method in Appendix A and the pipeline for training data construction in Appendix B. For training and evaluation, Section 4.1 specifies the large language models and datasets utilized, while Appendix C lists the corresponding configurations used. Furthermore, illustrative case studies are presented in Section D to offer qualitative insights into our method's application.

## ACKNOWLEDGEMENT

We would like to thank the anonymous reviewers for their constructive feedback, which greatly enhanced the rigor and clarity of our manuscript. Correspondence should be addressed to Shujian Huang and Shanbo Cheng. This work is supported by the following funding sources: the National Science Foundation of China (Grant No. 62376116); the Research Project of Nanjing University-China Mobile Joint Institute (Grant No. NJ20250038); the Fundamental Research Funds for the Central Universities (Grant No. 2024300507); and the Fundamental and Interdisciplinary Disciplines Breakthrough Plan of the Ministry of Education of China (Grant No. JYB2025XDXM118).

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

## A    CONSTRUCTION OF DUAL QUESTION FOR MATH REASONING

We propose a simple approach for construction of dual question for mathematical reasoning. The algorithm operates on mathematical expressions and performs the following key steps:

1. **Pattern Recognition and Exclusion:** The algorithm first identifies numerical candidates within the expression while excluding numbers in specific contexts: subscripts ($x_1, x_2$), inequality constraints ($x \leq 5$), common exponential bases ($2^n, 10^k$), and function arguments ($f(3)$).

2. **Variable Generation and Replacement:** For each valid numerical candidate, the system generates a unique variable identifier of the form $Variable_{\{str\}}$ where $str$ is a randomly generated lowercase string. The original number is then substituted with this variable.

3. **Question Generation of Dual Task:** Using the transformed expression and the original answer, the algorithm constructs inverse problems following templates such as: "Given that the correct answer is $\{answer\}$, determine the value of $\{variable\}$."

This methodology enables systematic generation of problem variants while preserving mathematical validity and semantic coherence. From a single primal question, multiple dual questions can be derived. To ensure that these dual questions robustly satisfy the properties of duality, we filter the candidates using the following two principles:

1. **Answerability of the Dual Question:** For the set of sampled answers collected for a given primal question, at least one answer must be capable of correctly solving the corresponding dual question.

2. **Uniqueness of the Correct Completion:** Among the same set of sampled answers, at most one should correctly answer the dual question.

Taken together, these two principles ensure that for any selected dual question, there is one and only one correct answer within the pool of candidate solutions for the primal task. This establishes the one-to-one correspondence necessary for generating a reliable self-supervised reward signal.

## B    MATH RL DATASET PREPARATION

Our dataset preparation process began with the collection of 1,815,942 prompts from various publicly available datasets (Chen et al., 2025; Albalak et al., 2025; He et al., 2025; Ji et al., 2025). After deduplication, we obtained 318,649 primal questions and generated 1,059,671 dual questions through our designed steps as discussed above. After that, we employed Qwen2.5-7B-Instruct (Yang et al., 2024a) to sample 32 candidate answers for each primal question and then prompted it to answer the corresponding dual question based on these candidates. Subsequently, we rigorously filtered out all dual questions that failed to meet our predefined principles above. We repeated this sampling and filtering process with Qwen3-4B (Yang et al., 2025), this time with 8 candidate answers per question. The resulting collection of high-quality, diverse mathematical questions along with corresponding dual questions formed our final RL training set, providing a robust foundation for our reinforcement learning process in the mathematical domain.

## C    EXPERIMENT DETAILS

We present more details about our training as follows. For the training process, we use a train batch size of 512, mini batch size of 32, sampling temperature of 1.0, and 16 rollouts per prompt. We employ the AdamW optimizer with a learning rate of 1e-6 and weight decay of 0.1 (following the default configuration in verl), with gradient clipping set to 1.0. For translation tasks, we set the maximum input length to 2,048 tokens and output length to 4,096 tokens. For mathematical tasks, we use the same input length but extend the maximum output length to 30,000 tokens.

## D    CASE STUDY

To illustrate the efficacy of our DuPO approach, we present two representative scenarios in Tab. 5 that demonstrate how DuPO provides a reliable reward signal across diverse domains.

**Scenario 1: Mathematical Reasoning Validation.** In mathematical reasoning, DuPO derives dual task questions from the primal task question where key numerical parameters are replaced with

| Scenario 1: DuPO on Mathematical Reasoning | |
|---|---|
| **Primal Task** | Let $\triangle ABC$ have circumcenter $O$ and incenter $I$ with $\overline{IA} \perp \overline{OI}$, circumradius **13**, and inradius **6**. Find $AB \cdot AC$. (Correct Answer: 468) |
| **Dual Task #1** | Let $\triangle ABC$ have circumcenter $O$ and incenter $I$ with $\overline{IA} \perp \overline{OI}$, circumradius $V_{sk}$, and inradius 6. Find $AB \cdot AC$. Check your work: If the solution for above question is $\boxed{answer}$, what must $V_{sk}$ have been? |
| **Dual Task #2** | Let's examine: Let $\triangle ABC$ have circumcenter $O$ and incenter $I$ with $\overline{IA} \perp \overline{OI}$, circumradius 13, and inradius $V_{rj}$. Find $AB \cdot AC$. When the solution for above question is $\boxed{answer}$, what's the corresponding $V_{rj}$? |
| **Candidates** | **Answer: 468    Backward Accuracy: 69.1%**
**Answer: 108    Backward Accuracy: 0%**
**Answer: 312    Backward Accuracy: 0%** |
| Scenario 2: DuPO on Machine Translation (MT) | |
| **Primal Task**

**Reference** | Translate to Chinese: As knowledge of Greek declined, the West found itself cut off from its Greek philosophical and scientific roots.
随着希腊知识的衰落，西方脱离了其希腊哲学和科学根源。 |
| **Primal MT #1**

**Dual MT #1** | 随着希腊语知识的衰落，西方发现自己与希腊的哲学和科学根源失去了联系。**(BLEU: 45.85)**
As knowledge of Greek declined, the West found itself cut off from its philosophical and scientific roots in Greece.**(BLEU: 82.07)** |
| **Primal MT #2**

**Dual MT #2** | 随着对希腊语的了解逐渐消失，西方发现自己与希腊哲学和科学根源隔绝开来。**(BLEU: 28.65)**
As understanding of the Greek language gradually fades, the West finds itself cut off from the roots of Greek philosophy and science.**(BLEU: 16.11)** |

Table 5: **Case Studies of DuPO on Mathematical Reasoning and Machine Translation.** DuPO validates each candidate's quality through a corresponding dual task, reliably identifies **the superior solution** over **inferior ones**.

variables, and the model tries to work backwards conditioned on candidate answers. When given a geometry problem about triangle properties, three candidate answers are sampled: 468, 108, and 312. DuPO automatically derives two dual questions by replacing the circumradius (13) and inradius (6) with variables, asking the model to deduce these values from the proposed answer. The candidate answer 468 achieves 69.1% accuracy on dual task, while the incorrect answers (108 and 312) totally fail to answer the dual task.

**Scenario 2: Machine Translation Quality Assessment.** For translation tasks, DuPO leverages reverse direction translation as the dual task to evaluate translation quality. Given an English sentence about Greek philosophical decline, two Chinese translation candidates are generated and subsequently back-translated to English. The first translation achieves a BLEU score of 45.85 in the forward direction and 82.07 in the back-translation, demonstrating semantic preservation and translation fidelity. In contrast, the second candidate shows degraded performance with BLEU scores of 28.65 and 16.11, respectively, indicating semantic drift and poor translation quality.

These case studies validate DuPO's core hypothesis: high-quality solutions maintain consistent information across dual task formulations, while inferior solutions exhibit significant degradation. This dual validation mechanism provides a robust framework for automatic quality assessment without requiring ground truth labels.

# E  LIMITATION DISCUSSION

DuPO has demonstrated its effectiveness on mathematical reasoning (representing complex reasoning tasks) and machine translation (representing constrained generation tasks), providing a novel

pathway for self-supervised training in LLMs. However, fully open-ended tasks remain inherently challenging. Tasks such as creative writing, where outputs are highly unconstrained and evaluation criteria are subjective, require further exploration to effectively extract the unknown component and construct meaningful dual problems. Intuitively, taking creative writing as an example, specific constraints in user instruction (e.g., theme, style, tone) might be extracted as the unknown component $x_u$ for reconstruction. However, the practical implementation and validation of such mechanisms in these open-ended domains constitute important directions for future work.

## F  THE USE OF LARGE LANGUAGE MODELS

We utilized a Large Language Model (LLM) as an assistive tool to proofread the manuscript for grammatical errors and to provide suggestions for improving clarity and flow. All suggestions were manually reviewed by the authors, who made the final decisions on all textual modifications to enhance the paper's readability.

