# OpenReview forum: "DuPO: Enabling Reliable Self-Verification via Dual Preference Optimization"
_ICLR.cc/2026/Conference — ICLR 2026 Poster_

### Official Review · Reviewer_P36R · 2025-10-17

**Soundness:** 3
**Presentation:** 3
**Contribution:** 3
**Rating:** 8
**Confidence:** 4

**Summary:**

This paper introduces a self-verification reward idea for reinforcement learning in LLMs. The general setting is given a prompt $x$, for the LLM to provide an answer $y$, as $\pi(y|x)$. The core idea of dual learning is to use the prediction of the prompt from the answer as a dual self-supervised training signal. The authors observe that it is not sufficient or possible for many tasks, identifying non-unique reconstruction or failure to reconstruct as core problems. To tackle this, they propose to decompose the prompt $x$ into  a known and an unknown parts, $x = (x_k, x_u)$, to use a model to predict the unknown part from both the answer and the known part, $\hat{x}_u = \mathcal{T}(y,x_k)$, and to use this to define a (self-verification) reward signal $r(x,y) \propto \exp(-\lambda d(x_u, \hat{x}_u))$, with $d$ being a task-dependent distance function. This reward is then optimized using RL. The proposed approach is then quite thoroughly experimented.

**Strengths:**

* The paper is very well written and structured, a pleasure to read
* The idea of decomposing the prompt for extending applicability of dual learning, and using it as self-verified reward signal is simple yet elegant and very interesting
* The experimental section is thorough and the results seem to be quite strong

**Weaknesses:**

* The approach is presented as being very general (with mentions to code generation, dialogue systems, open-ended and creative domains), while it is strongly dependent on the prompt decomposition which has to be defined manually for every task. Math reasoning and translation are the considered tasks, and only math reasoning calls for such a decomposition. This is fine, but this is a limitation that should be acknowledged and discussed.
* If the paper does a very good job at providing the high-level intuition, it somehow fails to provide a precise enough description of the proposed approach, harming understanding and reproducibility. See questions for more details.
* There are missing details harming reproducibility. The authors promise for open-sourcing the code, which is very nice, but some additional details could be provided in the appendix. See questions for more details.

**Questions:**

I'm supportive of accepting this paper, which is really great, provided that the questions below are answered, especially those regarding clarifying the proposed approach (which is probably a rather minor modification)

### More details on the proposed approach

* The paper currently doesn’t say what model is used for generating $x_u$ from $y$ and $x_k$. From the context, it is likely the LLM being trained, but is it the initial frozen LLM, the LLM being trained (as training progresses), or even another LLM?
* In all the above cases, the reward being optimized depends heavily on the capacity of the LLM to provide correct answer to the dual problem (predict $x_u$). If it’s not capable of making that prediction correctly enough, there is not learning signal. Can you comment on this? Notably, for the Llama experiment (table 3), the initial score is pretty low, suggesting that probably predicting $x_u$ may not be good enough, which makes the huge improvement quite surprising. Or is it that predicting $x_u$ is much easier than predicting $y$?
* Is there a mechanism for improving the prediction of $x_u$ (if the current LLM is used for building the reward function)? Or is it a byproduct of improving the prediction of $y$? This would be an interesting experiment to add, measuring the initial reward quality (given the ground truth $y$), and how it evolves along learning, once the precise setup has been clarified.

### Reproducibility details

Some parameters are not provided, such as the optimizer, learning rate and other related parameters, but more importantly the proposed reward function relies on hyper parameters that are not provided, and they should be provided but also ideally the sensitivity of the approach to their value should be studied. Notably, the choice of $\lambda$ and of the distance $d$.

### Other comments and questions

* What is called preference optimization (eg title of sec 3.3) is not preference optimization, it’s RL. It is semantics, but also misleading.
* Everywhere but in Eq (1), the reward does not depend on $x$, which is hopefully a typo
* A paper that would be interesting to discuss (no experiment needed) is “Reinforcement Learning Teachers of Test Time Scaling” by Sakana. They address a different problem, but there is some interesting similarity in the idea of decomposing the prompt and predicting part of it.
* Sec. 4.5.2 (scaling reasoning during inference without training) is very interesting. Please discuss the additional cost. Also, an experiment that would be quite interesting would be to run this on huge close-source strong models like the latest R1, through the API (as there is no learning), to see the amount of possible improvement there.

---

> ### Author Response · Authors · 2025-11-20
> **Response to Reviewer P36R**
>
> We sincerely thank the reviewer for the **strong endorsement** and for finding our work **"simple yet elegant"** and **"a pleasure to read."** We appreciate the constructive feedback on clarity and reproducibility. Below, we address the key questions.
>
> > Limitation Discussion
>
> We appreciate this insightful observation. We have added a "Limitations" section to the paper to discuss this point.
> We agree that fully open-ended tasks (e.g., Creative Writing) pose greater challenges and practical implementation requires further exploration.
>
> We emphasize that DuPO serves as a new pathway to unlock self-supervision training, demonstrated by impressive results in Mathematical Reasoning (representing complex reasoning tasks) and Machine Translation (representing constrained generation tasks). Intuitively, in Creative Writing, specific user constraints (e.g., theme, style, tone) might be extracted as the unknown component ($x_u$) to be reconstructed by the dual task, providing a possible solution for future exploration.
>
> > What model is used for generating x_u?   If the model is weak, is the signal reliable?
>
> We appreciate the reviewer's insightful question.
>
> * **Model Identity:** We clarify that the LLM being trained (current policy model) generates both the primal problem solution and the dual reconstruction during RL training.
>
> * **Reliability via Filtering:** After constructing dual questions via regex extraction and text templates, we conduct a rigorous **filtering process** (Appendix A). We utilize the initial LLM to conduct sampling on potential dual tasks, ensuring that there one and only one 'y'  that can correctly solve the dual question. Consequently,we discard dual questions that are "unsolvable" (model fails consistently) or "guessable" (multiple answers fit). As demonstrated in Figure 4, this **fully automated** pipeline significantly enhances the accuracy of the initial reward signal.
>
> * **Evolving during training:** As the model **improves during training**, it naturally becomes capable of solving more dual questions, **"unlocking" more** reward signals. Figure 3 confirms this positive loop: dual task accuracy rises in correlation with test set performance.
>
> > Reproducibility Details ( Paper Writing)
>
> We discussed the distance metric d in Line 214. Specifically, for mathematical reasoning, we evaluate the reconstruction of  using a binary 0/1 reward based on exact equality; for machine translation, we utilize the BLEU score.
>
> Following your suggestion, we have further updated Appendix C to include more details and apologize for the omission. We use the AdamW optimizer. The sensitivity parameter lambda is set to 1 by default.  We hope these updates will facilitate reproducibility and the broader application of our method.
>
> > Other comments and questions
>
> Thank you for bringing 'Reinforcement Learning Teachers' to our attention.  Its core idea, decoupling the reasoning process from the  solution to evaluate process quality based on student performance, to be highly interesting. We will include a discussion of this work in the Related Work section.
>
> For Inference Scaling: functioning as a **training-free reward model**, DuPO offers **flexible computational costs**. It supports scalable performance, ranging from efficient single-sample inference (approx. x1 cost, +6.9 Acc on AIME24, compared with Avg@32) to  **stronger System-2 style expansion** (N=8  as in paper,, approx. x8 cost with +16.6 Acc), requiring no parameter updates.
>
> Inspired by your suggestion,  we conducted experiments on the DeepSeek-R1-0528 (API). DuPO boosted R1's accuracy on AIME24 from 91.4% to **96.56%**. This result shows DuPO can further improve even the strongest model. We are completing the experiments for this suggested setting and will include them in the paper as soon as possible.

---

> ### Comment · Reviewer_P36R · 2025-11-25
> **Rebuttal acknowledgement**
>
> I acknowledge having read the rebuttal. I was positive about the paper, and the answer clarifies most of the points I raised. A few additional points though:
> * Following up on the point "evolving during training", in Fig 3, I can see the primal task accuracy (forward accuracy), but not the dual task accuracy?
> * I do not see the update in the related work section? I would trust the authors to add it later, but generally speaking it would make it easier to show more explicitly what changed in the revision, for example by putting the additional text in color.

---

> ### Author Response · Authors · 2025-11-27
>
> Thank you for your positive feedback and for engaging in this further discussion.
>
> 1.  Figure 3: We are updating the figure to explicitly include the dual task accuracy.
> 2.  We are incorporating the additional discussion into this section. We will ensure that all new text is highlighted in color to make the changes easy to track.
>
> We are currently finalizing the revision and we will upload the revised manuscript shortly.

---

### Official Review · Reviewer_tLFR · 2025-10-31

**Soundness:** 1
**Presentation:** 2
**Contribution:** 2
**Rating:** 2
**Confidence:** 4

**Summary:**

The paper proposes DuPO, a self-supervised preference-optimization framework that replaces external reward signals (human or verifiable answers) with a generalized duality signal. The idea is to decompose each input into known and unknown parts $(x_k, x_u)$ and define a dual task that reconstructs only the unknown parts from the model’s output and the known context. The reward is the distance between the true $x_u$ and its reconstruction. DuPO then optimizes the promal policy $\pi_{\theta}(y|x)$ with standard RL updates (GRPO in experiments). The authors conduct the experiments on translation, math reasoning, and inference reranking tasks.

**Strengths:**

* The idea of integrating dual learning into self-supervised LLM fine-tuning is relatively new and underexplored.
* The authors conduct experiments across several tasks, including both LLM fine-tuning and reranking.
* The paper includes a nice ablation study on component selection.

**Weaknesses:**

* Although the idea of dual learning is relatively new, the motivation for why it improves performance remains unclear. The notion of generalized duality is defined axiomatically, but there are no guarantees showing that optimizing the DuPO reward leads to better true task utility, nor theoretical bounds relating to the choice of decomposition. For tasks like machine translation, the intuition makes sense as the forward and backward tasks are naturally aligned, but for other tasks, it is unclear why DuPO would outperform existing methods such as RLVR or other self-play approaches.
* Several claims in the paper are problematic. For instance, the authors state that RLVR and RLHF rely on costly labels. However, constructing an appropriate dual task for DuPO may be even more challenging. Moreover, this may not even be possible for tasks such as creative writing. While the authors briefly mention the creative writing task, they do not discuss how it could be resolved.
* Reward design: It is unclear how the reward is defined when there are multiple valid outputs for the reconstructed unknown. For example, in a math problem with a single correct answer but many possible solution paths, how is the reward computed consistently?
* Baselines: The paper lacks RLVR baselines on the math reasoning task, as well as self-play LLM baselines such as [1-4].

[1] Chen, Zixiang, et al. "Self-play fine-tuning converts weak language models to strong language models." ICML 2024.

[2] Pang, Richard Yuanzhe, et al. "Iterative reasoning preference optimization." NeurIPS 2024.

[3] Dong, Qingxiu, et al. "Self-boosting large language models with synthetic preference data." ICLR 2025.

[4] Wu, Yue, et al. "Self-play preference optimization for language model alignment." ICLR 2025.

**Questions:**

* How should the inputs be decomposed into known and unknown parts in tasks such as creative writing?
* How does the choice of known and unknown parts affect the reward? For example, if the unknown parts are too simple, can the model predict them from the known parts alone?

---

> ### Author Response · Authors · 2025-11-20
> **Response to Reviewer tLFR  - part (1/2)**
>
> In response to the reviewer’s comments, we first restate the precise scope and contribution of DuPO.
>
> Our core contribution is a annotation-free paradigm that generates a **reliable self-verification reward** for LLMs via **generalized duality**. We decompose each input into known ($x_k$) and unknown ($x_u$) components and define a complementary dual task $T_cd$ that reconstructs only $x_u$ from the model’s output $y$ and the known context $x_k$. In math reasoning, $x_u$ is a variable of the input question and its value is **deterministic** and verifiable.  Furthermore, we filter $x_u$ under two principle Answerability and Uniquenes (Appendix A), ensuring one and only one y can reconstruct $x_u$, discarding those **'unsolvable'** and **'guessable'** dual question, making $x_u$’s reconstruction a reliable proxy for reward of $y$.
>
> DuPO offers a possible and promising pathway for self-verification training for various tasks by proposing generalized duality. Empirically, DuPO yields **consistent** gains across **two different and representative tasks** over various LLMs (e.g., avg +6.4 points on math reasoning, avg +2.1 COMET over 756 translation directions;) and even **tracks the upper bound baseline Oracle-RLVR** performance curve closely (Appendix E), achieving comparable accuracy without accessing any external ground truth.  We also acknowledge that further exploration and validation are needed for more open-ended settings such as creative writing.
>
> > W1  Why DuPO Works
>
> Regarding the concern about "guarantees of true task utility," we provide both structural and empirical evidence:
>
> * **Problem's Inherent Structure / Consistency Check:** The inherent relation in our complementary dual task making wrong 'y' fail to reconstruct x_u. Consider an equation A+B=C. If C is miscalculated, then A'=C-B cannot equal A.
> * **Rigorous Filtering:** As detailed in Appendix A, we conduct sampling from the policy model itself, ensuring one and only one answer can correctly answer the dual problem, and discarding those "guessed" or "unsolvable" dual problems. Therefore, a correct $y$ leads to a correct reconstruction of $x_u$, while an incorrect $y$ fails, ensuring the reconstruction acts as a clean and precise reward signal.
> * **Empirical Evidence:** We substantiate this claim with the Oracle-RLVR experiment (Figure 5), where DuPO tracks the performance trajectory of an oracle baseline (using ground-truth labels) almost exactly. This empirically proves that our self-supervised signal achieves the similar utility as perfect external supervision, validating DuPO’s effectiveness beyond intuition.
>
> > W2 Difficulty of Constructing an Appropriate Dual Task
>
> We respectfully disagree on the cost comparison. The critical distinction between DuPO and RLVR/RLHF is DuPO **doesn't require external supervision signal.** The construction of dual problems in DuPO is fully **automated**, making it both cost-effective and highly scalable, as evidenced by our experiments on two high-value and representative tasks: mathematical reasoning (for complex reasoning) and machine translation (for constrained-generation).

---

> ### Author Response · Authors · 2025-11-20
> **Response to Reviewer tLFR  - part (2/2)**
>
> > W3 How to Handle Multiple Valid Outputs for the Reconstructed Unknown
>
> We respectfully clarify a misunderstanding regarding the reconstruction target.
>
> In mathematical reasoning, $x_u$ represents the target variable in the problem statement to be reconstructed, whose value is **unique** (Case Study in Table5). Verifying the correctness of the reconstruction only requires matching methods to check if the final prediction is equal,  which is **simple and precise**. For different reasoning paths on dual problem that correctly predict $x_u$, each indicates that the corresponding forward-predicted $y$ is correct (as we also discussed in W1), and we assign a reward of 1 to all such cases.
>
> ---
>
> > W1/W4:  Baseline of RLVR/Self-Play
>
> We wish to clarify the misunderstanding as below:
>
> * **Without External Supervision**: RLHF, RLVR, and some self-play methods are valuable and effective but they might still rely on external signals (human annotations[1, 3, RLHF], ground-truth answer as verifiers [2, RLVR], or advanced reward model [4]). DuPO operates in a  self-supervised manner and derives rewards without external supervision.
>
> * **Direct Comparison with Oracle-RLVR**: In Table 3 We have already explicitly compared DuPO against RLVR baselines that utilize ground-truth labels. DuPO significantly outperforms SimpleRL-Zoo (32.1% vs. 19.0%). To further validate our method, we conduct Oracle-RLVR baseline (Appendix E) that uses ground-truth answers for verifying, representing the most accurate reward signal possible under the same dataset. DuPO tracks this oracle's performance closely, proving that our self-derived reward achieves comparable effectiveness as perfect external supervision
>
> * **Orthogonality**: Crucially, DuPO delivers significant gains across the various models, from weaker models (Base Model / LlaMA Model) to SOTA models already heavily optimized via various training paradigms (SFT and RLVR) (e.g.,Table 2). These improvements directly prove that DuPO is orthogonal to existing training paradigms and can be combined with them.
>
> ---
>
> > Q1/W2 Generalization to Board Domain (eg, Creative Writing)
>
> We agree that fully open-ended tasks, such as Creative Writing, are inherently challenges and require further exploration for practical implementation, a boundary we have explicitly discussed in our Limitations section.
>
> Nevertheless, we emphasize that DuPO establishes a novel and promissing pathway for **unlocking self-supervision**. Our reported results on Mathematical Reasoning and Machine Translation have demonstrated the method's efficacy across **two representative and high-value tasks** of current LLMs.
>
> Furthermore, DuPO offers a possible pathway for tasks previously deemed non-invertible. Intuitively, take creative writing as am example, specific user constraints (e.g., theme, style, tone) can be modeled as the unknown component  to be reconstructed by the dual task, providing a direction for future research.
>
> ---
>
> > Q2  Risk of Dual Problem Design
>
> Thank you for your critical question. As discussed in W1, we have conducted the unknown component selection strategy ensuring that there is a unique $y$ capable of correctly reconstructing $x_u$. This significantly mitigates the scenarios you concerned. Furthermore, the ablation study in Figure 4 empirically validates the effectiveness of this strategy.

---

> ### Author Response · Authors · 2025-11-23
>
> We hope that our previous response and the updated revision have effectively addressed your concerns. We remain fully available for any further follow-up questions or clarifications

---

> > ### Comment · Reviewer_tLFR · 2025-11-26
> >
> > Thank you to the authors for their response, which addressed most of my concerns. Therefore, I am raising my score to 6.
> >
> > I still believe the paper requires substantial revision, particularly in the introduction to better highlight its novelty and contributions and to avoid overclaiming. For example, the authors mention open-ended tasks as a limitation for RLVR, yet the paper does not fully resolve this issue. In the introduction, the authors mentioned creative writing as a task that lacks strict invertibility, but the paper does not actually address this problem in the creative-writing domain nor provide empirical validation, which may cause confusion.

---

> > > ### Author Response · Authors · 2025-11-27
> > >
> > > We sincerely thank the reviewer for the positive feedback and for raising the score. We are currently finalizing further revisions based on your constructive feedback and will upload the updated paper very soon.

---

### Official Review · Reviewer_EbeA · 2025-10-31

**Soundness:** 2
**Presentation:** 3
**Contribution:** 3
**Rating:** 4
**Confidence:** 4

**Summary:**

This paper proposes DuPO (Dual Learning-based Preference Optimization), a novel framework for optimizing Large Language Models (LLMs) without external supervision. DuPO addresses the limitations of traditional dual learning—which requires strictly invertible task pairs—and Reinforcement Learning with Verifiable Rewards (RLVR)—which relies on costly ground-truth labels. The core innovation is a generalized duality framework: it decomposes the input of a primal task into known and unknown components, and then defines a dual task that reconstructs only the unknown component using the primal output and the known input. The quality of this reconstruction serves as a self-supervised reward signal to optimize the primal task. The authors demonstrate DuPO's effectiveness on both mathematical reasoning and multilingual translation, showing significant performance gains during training and as a training-free inference-time reranker.

**Strengths:**

1. Fully Self-Supervised: The method requires no human labels, reference answers, or external reward models, which is a major advantage for scalability and cost-efficiency.
2. Practical Impact: The ability to use DuPO as a simple, training-free reranker that boosts performance by +9.3 points is a highly practical contribution.

**Weaknesses:**

1. Insufficient experiments: In most experiments, DuPO is compared to untrained models, which is unfair. For translation tasks, RLVR may be difficult, but it is still worth using as a baseline. For mathematical tasks, standard RLVR should be added as a baseline. In addition, SFT can be added to each benchmark as a baseline.
2. Task Construction: While the paper provides principles for selecting the unknown component (Appendix A), the process still seems somewhat heuristic and task-specific. I still have concerns about the generality of this method on other tasks.

**Questions:**

N/A

---

> ### Author Response · Authors · 2025-11-20
> **Response to Reviewer EbeA**
>
> We appreciate the reviewer's recognition of DuPO as a **scalable, self-supervised** framework with **significant practical value**. We address the concerns regarding baselines and generality below.
>
> > W1: Insufficient Experiments & Baselines ( SFT / RLVR ).
>
> We respectfully point out a misunderstanding regarding our experimental setup:
>
> * In Table 3, we already compare DuPO against SimpleRL-Zoo (a standard RLVR implementation), where DuPO achieves a significant margin (**32.1% vs. 19.0%**). Inspired by your suggestion and to further validate our method, we strictly use the same dataset and conduct Oracle-RLVR baseline (Appendix E) that uses **ground-truth annotation**, representing the **most accurate** reward signal. DuPO **tracks this oracle's performance closely**, proving that our self-derived reward achieves comparable effectiveness as perfect external supervision
>
> * Meanwhile, it is worth noting that all models reported in Tables 1 & 2 (eg,. DeepSeek-R1, Nemotron-7B) have already undergone various training paradigms (eg,. SFT/RLVR) extensively. DuPO **consistently delivers further performance** gains on top of these strong model empirically demonstrates that our method is **orthogonal** to existing supervised paradigms and could be **combined** with.
>
> > W2: Concerns about Generality.
>
> We clarify that DuPO is governed by a unified design: decomposing inputs to construct complementary dual tasks where reconstruction serves as a rigorous reward. This mechanism is successfully validated across the **two critical and representative**  LLM tasks: Complex Reasoning (Math) and Constrained Generation (Translation).
>
> Regarding open-ended tasks (e.g., Creative Writing), while we acknowledge the challenges and further exploration needed (discussed in Limitations Section), DuPO offers a new possible pathway for **extending self-supervision** to broader applications. Intuitively, taking creative writing as an example, DuPO might extract specific constraints (e.g., theme, style) as the unknown component  for reconstruction.

---

> ### Author Response · Authors · 2025-11-23
>
> We hope that our previous response and the updated revision have effectively addressed your concerns. We are open to further discussion should you have any remaining concerns.

---

> ### Comment · Reviewer_EbeA · 2025-11-26
>
> Thank you for the authors' responses. It is indeed difficult to obtain accurate reward signals in general tasks, and I understand this.
>
> Regarding the baseline issue, although most models are trained with SFT/RL, such as DeepSeek-R1 and Nemotron-7B, the training data are not the same.
> For example, in Table 2, although it indicates that DuPO has improved by 6.5% on the basis of Qwen3-4B, this number cannot accurately give me feedback on whether DuPO it is good or bad. I hope there are at least 1-2 baselines in this table (using the same data, even labeled data, as Oracle results) that can be used for comparison.
>
> Also, thank you for providing the results in Appendix E. If the issue with Table 2 is not fully resolved, you can provide me with partial results before the deadline, and I will also consider increasing my score before the deadline.

---

> > ### Author Response · Authors · 2025-11-26
> > **Response to Reviewer EbeA**
> >
> > Thank you for your constructive feedback and for considering improving the score. We agree that a controlled baseline using the exact same training data is crucial to isolate the effectiveness of the DuPO.
> >
> > ---
> >
> > To address this, as you noted, we have performed the controlled experiments (Appendix E) using Qwen3-4B-Base, acting as the fair, controlled comparisons:
> >
> > * **Oracle-RLVR (requested)**: This baseline uses the **exact same training data** as DuPO. The only difference is that Oracle-RLVR utilizes **annotated ground truth** (standard answers) for verification to provide the **Oracle Reward**. Since this reward is the most accurate, this baseline serves as the upper bound for RLVR performance.
> > * **Oracle-DAPO**: We also replicated the experiments using data from DAPO (representative of **high-quality open-source data**) along with its ground truth annotations.
> >
> >
> > The experimental results demonstrate that DuPO’s performance is very close to both Oracle baselines. This directly validates the accuracy of the DuPO reward and the effectiveness of our method.
> >
> > ---
> >
> > We are actively conducting additional experiments (across different base models as in Table2) and revising the manuscript (including merging these results into Table 2 as suggested).
> >
> > We welcome further discussion to address any remaining concerns you may have.

---

### Official Review · Reviewer_S26D · 2025-11-03

**Soundness:** 3
**Presentation:** 3
**Contribution:** 3
**Rating:** 6
**Confidence:** 3

**Summary:**

The authors introduce DuPO, a consistency-based preference optimization framework which exploits task duality to increase post-training robustness. The authors achieve success over existing baselines on math datasets, and extend their framework to work at inference-time as well.

**Strengths:**

- The paper is well-written, easy to follow, and the figures are detailed with the appendices including relevant information and concrete examples
- The idea of using dual problems in RLHF is interesting and (to my knowledge) novel
- The experimental results appear rather promising

**Weaknesses:**

- My primary concern with DuPO is regarding the difficulty of constructing the dual problems, verifying the correctness of those duals problems, and how generalizable it is to different domains.
- In addition, it seems rather expensive to have to run DuPO at inference-time since the dual problems would also need to be generated on the fly?

**Questions:**

- Please address weaknesses

---

> ### Author Response · Authors · 2025-11-20
> **Response to Reviewer S26D**
>
> We thank the reviewer for acknowledging the **novelty of our approach** and finding the **experimental results promising**. We appreciate your constructive feedback regarding practicality and inference cost.
>
> > W1: Difficulty of Constructing/Verifying Dual Problems and Generalizability
>
> We clarify that our pipeline is fully **automated, scalable, and annotation-free**.
>
> * **Automated Construction:** Constructing dual tasks relies on simple rule-based extraction and text templates (see cases in Table 5) . This low-cost approach allowed us to easily scale to ~1M dual instances (Appendix B).
>
> * **Verification via Self-Filtering**: We do not need external supervision to verify formatted dual problems. Instead, we employ two princeple (Appendix A) by sampling from the policy model itself. A dual problem is retained only if the model can solve it uniquely and correctly based on one primal output. This automatically discards "unsolvable" or "guessed" noise, ensuring high-quality reward signals without human intervention.
>
> DuPO serves as a novel step to unlock self-supervision:
>
> * **Verified on Two Representative High-Value Domains:** We have verified DuPO on Mathematical Reasoning and Machine Translation, two representive and high value task for complex reasoning and constrained generation.
>
> * **Potential for Asymmetric Tasks:** As the reviewer correctly noted, open-ended domains like Creative Writing are challenging. However, the core mechanism of DuPO offers a new pathway for tasks previously considered non-invertible. Intuitively, even in Creative Writing, specific user constraints (e.g., theme, style, tone) can be extracted as the unknown component ($x_u$) to be reconstructed by the dual task, providing a foundation for future exploration.
>
> We acknowledge that fully open-ended tasks (e.g., Creative Writing) pose greater challenges and practical implementation requires further exploration. We have discussed it in the updated limitations section.
>
> > W2: Inference-time-Scaling Cost
> * **Zero Additional Inference Cost for Training Mode:** DuPO is primarily a post-training framework. Once the model is trained using DuPO (as shown in our main experiments), it achieves significant gains (e.g., +6.5% on Qwen3-4B) with no additional inference cost compared to the base model.
> * **Training-Free & Flexible Cost for Inference Mode:** DuPO further functions as a **training-free Reward Model**, optionally enabling a **System-2 style inference expansion** to leverage more compute for performance improvements **without training** model. The computational cost is highly configurable based on the user's budget: for high-resources scenarios, one can invest more computations (e.g.,  N=8 as in our paper) to obtain robust reward estimates (approx. x9 cost for **+16.6** Acc improvement on AIME24, compared with Avg32). Under constrained budgets, one can sample just a single response per dual question (approx. x1 cost for **+6.9** Acc improvement ).
>
> We thank the reviewer for this valuable suggestion. We are incorporating this discussion into the our manuscript.

---

> > ### Author Response · Authors · 2025-11-23
> >
> > We hope that our revisions and explanations have addressed your concerns. We are happy to engage in further discussion to ensure all concerns are met.

---

> ### Author Response · Authors · 2025-11-26
>
> We would like to inquire whether our response has successfully resolved your concerns. If your concerns have been resolved, we would be grateful if you would consider improving your score. Your feedback is highly valuable to us.

---

### Author Response · Authors · 2025-11-20
**General response**

> Q1: Why does DuPO work? (Soundness & Reliability)

DuPO leverages **Generalized Duality** to generate precise self-supervised reward signals **without external supervision**.

**Mechanism**: We decompose the input $x$ into known ($x_k$) and unknown ($x_u$) components. The dual task reconstructs $x_u$ using the model's output $y$ and $x_k$, where the value of $x_u$ is deterministic and verifiable.  Since this reconstruction relies on $y$, it serves as a metric for $y$'s quality, optimizing this reward signal enhances model performance.

**Reliability Assurance**:
* **Complementary Dual Task Structure:** The reconstruction relies on rigid logical constraints (e.g., in A+B=C, if output C is wrong, the reconstructed A' = C-B cannot match the original A).

* **Rigorous Filtering:**  As detailed in Appendix A, we conduct sampling from the initial LLM, ensuring one and only one answer can correctly answer the dual problem, and discarding those "guessed" or "unsolvable" dual problems. Therefore, a correct $y$ leads to a correct reconstruction of $x_u$, while an incorrect $y$ fails, ensuring the reconstruction acts as a precise reward.

* **Empirical Validation:** We conduct an *Oracle-RLVR* baseline (Appendix E) that uses ground-truth labels for rewards. DuPO tracks the Oracle-RLVR performance curve closely, achieving comparable accuracy without accessing any ground truth. This empirically proves that DuPO’s intrinsic reward is as reliable as external supervision.

---

>  Q2: Efficiency &  Difficulty of Constructing Dual Problem

DuPO offers a superior **highly efficiency** training paradigm compared to RLHF/RLVR:

* **Automatic Pipeline:** Take math reasoning as an example, constructing dual tasks requires only rule-based extraction and text templates, enabling the generation of massive dual problems with minimal engineering effort. Furthermore, the filtering strategy relies solely on sampling from the policy model itself, improving reward without requiring external advanced reward models or human labelers.

* **Scalability:** We successfully scaled this process to ~318k primal questions (generating ~1M dual instances), validating its robustness and low overhead on large-scale datasets (Appendix B).

---

> Q3: Generalization & Scope

We emphasize that DuPO serves as a novel pathway to unlock self-supervision training:

* **Verified on Representative High-Value Domains:** We demonstrate impressive results in Mathematical Reasoning (representing complex reasoning tasks) and Machine Translation (representing constrained generation tasks). These two domains cover the critical "reasoning & generation" capabilities of current LLMs.

* **A Blueprint for Open-Ended Tasks:** While fully open-ended tasks (e.g., Creative Writing) are inherently challenging, DuPO offers a novel pathway. Intuitively, in creative writing, specific user constraints (e.g., theme, style, tone) can be extracted as the unknown component ($x_u$) to be reconstructed by the dual task, providing a foundation for future exploration.
Future Direction: We explicitly discussed the challenges of open-ended domains in the updated Limitations section. However, given the strong performance in math reasoning and transaltion, DuPO stands as a valid and scalable step towards generalized self-verification.


---

> Q4: Baseline Comparison

We clarify our position regarding baselines:

* **Direct Comparison with Oracle-RLVR:** We explicitly compare DuPO against RLVR baselines that utilize ground-truth labels. In Table 3, DuPO significantly outperforms SimpleRL-Zoo (**32.1% vs. 19.0%**).

* **Approaching the Oracle:** To further validate our method, we strictly use the same dataset and conduct **Oracle-RLVR** baseline (Appendix E) that uses ground-truth labels representing the most accurate reward signal. DuPO tracks this oracle's performance closely, proving that our self-derived reward achieves comparable effectiveness as perfect external supervision

* **Without External Supervision:** RLHF, RLVR, and part of self-play methods are valuable and effective but they might still rely on external signals (human annotations, ground-truth answer as verifiers, or advanced reward model). DuPO operates in a self-supervised manner and derives rewards solely from intrinsic task duality.

* **Robustness & Orthogonality:** Crucially, DuPO delivers significant gains across the various models, from weaker models (Base Model / LlaMA Model) to **SOTA models** already heavily optimized via various training paradigms (SFT, RLVR) (e.g.,Table 2). These improvements empirically prove that DuPO is **orthogonal** to existing training paradigms and can be **effectively combined** with them.

---

### Author Response · Authors · 2025-12-03
**Summary of the rebuttal status before the bug-induced information leak:**

# **Summary of the rebuttal status before the bug-induced information leak:**
Rebuttal successfully addressed major initial concerns. All reviewers had expressed positive signals.

| Reviewer | Init Score | Final Score (Pre-Bug) | Key Action |
| :--- | :--- | :--- | :--- |
| **P36R**| 8 | 8 (Strong Support) | Validated on DeepSeek-R1 (+5.1% Acc) |
| **tLFR** | 2 | 6 (Concern Addressed) | Resolved misunderstanding & Added Oracle Baseline |
| **EbeA** | 4 | Willing to Raise(Confirmed)  | Provided controlled baselines (Oracle-RLVR) |
| **S26D** | 6 | 6 (Absence) | Addressed concerns; Reviewer absent |

---

## Reviewer Status：
* Reviewer tLFR: **2 → 6**
   - Reason:  Misunderstandings regarding method and baselines were resolved. We added an Oracle-RLVR experiment (Appendix E), directly demonstrating that DuPO achieves performance comparable to the Oracle baseline without any annotations, and we further clarified DuPO's theoretical mechanism. After the score was raised, we made further revisions to the paper to resolve the remaining concerns.
   - Quote："Thank you to ..., which **addressed most of my concerns**. Therefore, I am **raising** my score to 6."
* Reviewer EbeA: **4 →  Positive**
   - Reason: Provided requested controlled baselines (Oracle-RLVR/DAPO).
   - Quote: "consider **increasing my score**... **Thank you for providing the results** in Appendix E."
* Reviewer P36R: **8 (Maintain Strong Support)**
   - We engaged in a positive discussion with the reviewer and, following their suggestion, further validated the effectiveness of DuPO on DeepSeek-R1 (AIME24 score increased from 91.4% to 96.56%).
   - Quote: "the answer **clarifies most of the points I raised**."  "I was **positive about the paper**"
* Reviewer S26D: **6 (Absent)**
   - We addressed the reviewer's concerns by clarifying that the dual problem construction is annotation-free and fully automated, and provided a cost analysis for the inference-time-scaling. However, the reviewer was absent throughout the rebuttal phase.

---

## Consensus on Strength
* **Novelty & Impact**："simple yet **elegant** and very interesting" (P36R)，"**interesting** and **novel**" (S26D)，"new and **underexplored**" (tLFR)，"advantage for **scalability** and cost-efficiency"（EbeA）
* **Experimental Rigor**："experimental section is **thorough**" (P36R), "results appear rather **promising**" "**highly practical contribution**." (S26D),  "**nice** ablation study" (tLFR)

---

Beyond addressing the reviewers' concerns in our rebuttal, we also proactively incorporated their suggestions to update the manuscript. We sincerely thank the reviewers for their time and valuable suggestions.

---

### Meta-Review · Area_Chair_Q2US · 2026-01-11

**Summary:**

This paper was reviewed by four experts, receiving initial scores of 2, 4, 6, and 8. The rebuttal proved to be very effective; notably, the reviewer who initially gave a 2 increased their score to 6, and the reviewer with a 4 also expressed willingness to raise their rating. As the authors have satisfactorily addressed most of the concerns, the decision is to recommend acceptance. For the final version of the paper, the authors are expected to incorporate the remaining feedback to further strengthen the work. Specifically, this includes adding more in-depth discussion and analysis (as requested by Reviewers S26D and tLFR), providing more detailed experimental comparisons and ablation studies (Reviewers EbeA and tLFR), and refining the presentation to clarify the methodological insight and motivation (Reviewer P36R). The authors are encouraged to implement these changes to the best of their ability.

**Reviewer Concerns:**

Through the rebuttal process, the authors have effectively resolved several key concerns regarding the design motivation, paper presentation, and experimental comparisons. However, the manuscript would benefit significantly from further elaboration on the analysis and discussion. Providing deeper insights into these areas would strengthen the paper's arguments and make the final revised version more convincing.

**Reviewer Scores:**

The reviewers may acknowledge and appreciate the authors' efforts to improve the manuscript, specifically noting the enhanced presentation, clarified motivation, and additional comparisons. However, despite these improvements, concerns remain regarding the depth of the analysis. The consensus is that the discussion must be expanded to fully validate the findings and address the complexities of the proposed approach.

---

### Decision · Program_Chairs · 2026-01-26

Accept (Poster)